# **Technical Note: Calculating state dependent equilibrium climate sensitivity from palaeodata**

Peter Köhler<sup>1</sup>, Lennert. B. Stap<sup>2</sup>, Anna S. von der Heydt<sup>2</sup>, Bas de Boer<sup>3</sup>, and Roderik S. W. van de Wal<sup>2</sup> <sup>1</sup>Alfred-Wegener-Institut Helmholtz-Zentrum für Polar-und Meeresforschung (AWI), P.O. Box 12 01 61, 27515 Bremerhaven, Germany <sup>2</sup>Institute for Marine and Atmospheric research Utrecht (IMAU), Utrecht University, Princetonplein 5, 3584 CC Utrecht, The Netherlands

<sup>3</sup>School of Earth and Environment, University of Leeds, Leeds, UK

Correspondence to: Peter Köhler (Peter.Koehler@awi.de)

#### Abstract.

The evidence from both data and models indicate that specific equilibrium climate sensitivity  $S_{[X]}$  — the global annual mean surface temperature change ( $\Delta T_g$ ) as a response to a change in radiative forcing X ( $\Delta R_{[X]}$ ) — is state dependent. Such a state dependency implies that the best fit in the scatter plot of  $\Delta T_g$  versus  $\Delta R_{[X]}$  is not a linear regression, but for instance a higher

order polynomial. While for the conventional linear case the slope (gradient) of the regression is correctly interpreted as the specific equilibrium climate sensitivity  $S_{[X]}$ , the interpretation is not straightforward in the non-linear case. We here elaborate how such a state dependent scatter plot needs to be interpreted, and provide a theoretical understanding how to calculate  $S_{[X]}$  in the non-linear case.

### 1 Introduction

One prominent approach to calculate (specific) equilibrium climate sensitivity  $S_{[X]}$  from palaeodata is to evaluate the regression of scatter plots, in which global mean surface temperature change ( $\Delta T_g$ ) has been plotted against radiative forcing changes ( $\Delta R_{[X]}$ ), since following its definition,

$$S_{[X]} = \frac{\Delta T_g}{\Delta R_{[X]}},\tag{1}$$

S<sub>[X]</sub> is easily obtained from the slope of a linear regression line, that needs to pass through the origin to avoid any biases.
Passing through the origin (ΔT<sub>g</sub> = 0 K; ΔR<sub>[X]</sub> = 0 W m<sup>2</sup>) implies that no temperature change (with respect to a defined reference climate state) is detected for conditions without forcing anomalies. Usually the pre-industrial climate state serves as reference. Here, X corresponds to the forcing processes considered, typically changes in CO<sub>2</sub> (sometimes also including the other greenhouse gases CH<sub>4</sub> and N<sub>2</sub>O), potentially corrected for some slow feedbacks such as planetary albedo changes causes by variations in land ice (LI), vegetation (VE) or dust (aerosols (AE)). This nomenclature above follows the suggestion of a

20 recent review on this topic (PALAEOSENS-Project Members, 2012).

In the review of the PALAEOSENS group in 2012 a quantitative expression of  $S_{[X]}$  based on Equation 1 was already included, but only for individual data points, or whole time series, and a state dependent character of  $S_{[X]}$  was already detected, but could not be quantified in greater detail. Since then climate sensitivity from palaeodata has continued to be analysed by some regression analysis in the scatter plot of  $\Delta T_g$  versus  $\Delta R_{[X]}$  (e.g. von der Heydt et al., 2014; Martínez-Botí et al., 2015; Köhler et al., 2015).

- The analysis of the problem is straightforward, if linear regression methods are applied, which implies that  $S_{[X]}$  is a general property of the climate system. However, results point more and more in the direction, that climate sensitivity is state dependent, implying that for the  $\Delta T_g$ - $\Delta R_{[X]}$ -scatter plot non-linear regressions (for example higher order polynomials) are describing the data more appropriate than a simple linear fit (von der Heydt et al., 2014; Köhler et al., 2015). In such cases the quantification
- of  $S_{[X]}$  becomes more intricate.

The aim of this technical note is to briefly highlight potential pitfalls when analysing  $S_{[X]}$  from a data set, that indicates a state dependent behaviour, and offer understanding and solutions. These simple, but essential thoughts were not contained in our most recent contribution (Köhler et al., 2015). The key point of Köhler et al. (2015) was to show that given the available data climate sensitivity during the last 2.1 Myr is state dependent. One of the possible ways to quantify this state dependency is to

15 provide a probability density function or PDF (approach I, contained in subsection 2.1). Here we generalise different methods to quantify climate sensitivity in case it is state dependent. Hence this technical note offers a more general understanding of state dependent climate sensitivity based on palaeodata.

#### 2 Explaining different methods

We will in the following briefly expand on different ways how data can be analysed. Approaches are compared for the most
simplistic case of two data points only (Fig. 1), but we will for illustrative purposes also apply the different methods to the ice core data of the last 800 kyr already investigated in detail in Köhler et al. (2015). Finally, we also also include data of the last 2.1 Myr based on only CO<sub>2</sub> proxy data from the Hönisch-lab (Hönisch et al., 2009) in our conclusions. Please refer to Köhler et al. (2015) for the origin of the data and details of the applied statistical methods by which the data have been analysed.

The data of  $\Delta T_{\rm g}$  and  $\Delta R_{\rm [CO_2,LI]}$  based on ice core CO<sub>2</sub> and own model-based deconvolution of benthic  $\delta^{18}$ O into ice

- sheet albedo (radiative forcing) and global temperature as calculated in Köhler et al. (2015) are plotted in Figs. 2a,b. They include uncertainty estimates in both variables ( $\sigma_{\Delta T_g}$ ,  $\sigma_{\Delta R_{[CO_2,LI]}}$ ), uncertainties in the age models, however, have not been investigated. The temporally higher resolved CO<sub>2</sub> data have been averaged (resampled) to the temporal availability of the land ice sheet and temperature reconstruction of 2 kyr, leading to 394 data points almost equidistantly covering the last 0.8 Myr. Using a Monte-Carlo approach and *F*-test statistics a 3rd-order polynomial has been identified to best fit the scattered data of
- 30  $\Delta T_{\rm g}$  against  $\Delta R_{\rm [CO_2,LI]}$  (Fig. 2c).

#### 2.1 Approach I: The point-wise approach

The easiest and most robust estimate of  $S_{[CO_2,LI]}$  can be obtained, when  $S_{[CO_2,LI]}$  is calculated individually for each time step  $t_i$  out of  $\Delta T_g(t_i)$  and  $\Delta R_{[CO_2,LI]}(t_i)$  (Fig. 2d). Taking the uncertainties of the individual data points into account a probability density functions (PDF) of  $S_{[CO_2,LI]}$  is calculated straightforward (Fig. 2f), from which the median, the most likely and the spread (uncertainty distribution) within  $S_{[CO_2,LI]}$  can be obtained. If the underlying data set of the PDF is split into various subsets (here distinguishing data for two different radiative forcing domains, Fig. 2e) a first, rough quantification of the state dependency of  $S_{[CO_2,LI]}$  is generated, and has already been obtained in Köhler et al. (2015). One known problem of this approach is that for small disturbances in the radiative forcing ( $\Delta R_{[CO_2,LI]}$  close to zero) one might obtain in the pointwise calculations of  $S_{[CO_2,LI]}$  unrealistically high and low values. Such values might be caused by dating uncertainties of the

10 underlying palaeorecords or transient effects (de Boer et al., 2012). In our analysis in Köhler et al. (2015) we found 20 outliers (from 394 data points contained in Fig. 2) that did not match in the realistic range of S<sub>[CO2,LI]</sub> between 0 and 3 K W<sup>-1</sup> m<sup>2</sup>, and they have been rejected from further analysis. Furthermore, from those data with ΔR<sub>[CO2,LI]</sub> close to zero, that have not been rejected, calculated values of S<sub>[CO2,LI]</sub> seemed to follow a different pattern than the rest of the data (Fig 2e). Again, we understand these anomalies to be probably based on dating uncertainties, non-negligible influence of transient climate response and the problem that the ratio from two small numbers might easily contain a large error.

A state dependent equation of the specific equilibrium climate sensitivity  $S_{[X]}$  might be obtained from analysing the palaeodata in greater detail. To do so, a function has to be found that relates the temperature perturbations  $\Delta T_g$  to radiation perturbations  $\Delta R_{[X]}$ . For reasons of simplicity both variables are described in the following by  $\Delta T$  and  $\Delta R$ . For non-linear description of  $\Delta T$  as a function of  $\Delta R$  a higher order polynomial is the most obvious choice (but other equations are possible):

20 
$$\Delta T(=f(\Delta R)) = a + b\Delta R + c\Delta R^2 + d\Delta R^3 + \dots$$
 (2)

Climate sensitivity can then be calculated as:

$$S^{\rm pw} = \frac{\Delta T}{\Delta R} = \frac{a}{\Delta R} + b + c\Delta R + d\Delta R^2 + \dots$$
(3)

This approach is called *point-wise* (pw) since the derived equation in the S – ΔR data space agrees with the individual data points (Figs. 2e, 3b). The reference climate has to be chosen such that a = 0, to ensure finite climate sensitivity at ΔR = 0. In
the above case, climate sensitivity is constant (i.e. not state dependent) if the higher order terms are zero: c = d = 0. Otherwise, climate sensitivity is function of the radiation perturbation and therefore state dependent. This approach has been applied for the ice core data and is contained in Fig. 2e.

#### 2.2 Approach II: Using local slopes (piece-wise linear analysis)

In the constant case (c = d = 0), climate sensitivity can also be found by taking the *local slope* of the function, therefore called 30 S<sup>local</sup>:

$$S^{\text{local}} = \frac{\delta \Delta T}{\delta \Delta R} = b. \tag{4}$$

5

However, in the state dependent case:

$$S^{\text{local}} = \frac{\delta \Delta T}{\delta \Delta R} = b + 2c\Delta R + 3d\Delta R^2 + \dots$$
(5)

Now  $S^{\text{local}}$  (Equation 5) is evidently not equal to the point-wise-calculated climate sensitivity  $S^{\text{pw}}$  (Equation 3). For illustrative purposes we have included some realisation of S based on local slopes in Fig. 3b. Clearly, they disagree from results obtained with the point-wise approach. Indeed, the suggested equations do not meet the scattered data of  $S_{[\text{CO}_2,\text{LI}]}$  versus  $\Delta R_{[\text{CO}_2,\text{LI}]}$ .

The condition for state dependency, however, remains the same: the higher order terms have to be non-zero. For the local slope case, this means that the slope is non-constant.

## 2.3 Combining data-based approaches and model results

Climate models usually perturb a reference climate  $\{R_0; T_0\}$ , and end up with a new climate  $\{R_1; T_1\}$ . They consider climate sensitivity in the following way:

$$S^{\text{model}} = \frac{T_1 - T_0}{R_1 - R_0}.$$
(6)

One might argue, that only radiative forcing anomalies are of interest, and not the absolute radiative forcings  $R_0$  and  $R_1$ , so the denominator in Equation 6 should be  $\Delta R_1$ . For the sake of generalisation we keep Equation 6 as is, but the reader will see below (Equation 9) how relevant this formation might be.

Equation 2, the higher order polynomial fit of temperature versus forcing, is also valid, if based on absolute values in T and R, so one might fit in a model-output T versus R scatter plot:

$$T(=f(R)) = a + bR + cR^2 + dR^3 + \dots$$
(7)

Using this approach would imply (using Eq. 7, here reduced for simplicity to a 2nd order polynomial):

$$S^{\text{model}} = \frac{b(R_1 - R_0) + c(R_1^2 - R_0^2)}{R_1 - R_0}.$$
(8)

Now, if the reference climate  $(T_0, R_0)$  happens to be the pre-industrial reference climate (which is not always the case), we have  $\Delta R_1 = R_1 - R_0$ ;  $\Delta T_1 = T_1 - T_0$ , then:

$$S^{\text{model}} = \frac{b\Delta R_1 + c\Delta R_1^2 + 2cR_0\Delta R_1}{\Delta R_1} = (b + 2cR_0) + c\Delta R_1 = b' + c\Delta R_1, \tag{9}$$

with  $b' = (b + 2cR_0)$ . Equation 9 is equal to approach I, the point-wise quantification of climate sensitivity ( $S^{pw}$ ), but the parameter b' of the non-linear regression is depending on the reference climate, in detail the radiative forcing  $R_0$ .

More generally  $S^{\text{model}}$  can be obtained from  $S^{\text{local}}$ , by the following equation:

$$S^{\text{model}} = \frac{1}{R_2 - R_1} \int_{R_1}^{R_2} S^{\text{local}} d\Delta R.$$
(10)

$$= \frac{1}{\Delta R_2 - \Delta R_1} \int_{\Delta R_1}^{\Delta R_2} S^{\text{local}} \, \mathrm{d}\Delta R, \tag{11}$$

with  $\Delta R_i = R_i - R_0$ . This equation is generally valid, and agrees in case of the reference climate being the pre-industrial periods with  $S^{\text{pw}}$  (approach I). In other words, point-wise climate sensitivity is a measure of the mean local slope climate sensitivity over the radiation perturbation interval.

The benefit of using this local slope approach is that, in contrast to the point-wise approach, the sensitivity is not related to an arbitrarily chosen reference climate (often taken to be pre-industrial climate); it represents the slope of temperature as a function of radiation perturbations at each point.  $S^{\text{local}}$  is therefore more readily comparable to climate model results. In using this approach though, one has to realise that in the state dependent case  $S^{\text{local}}$  is not equal to the point-wise climate sensitivity  $S^{\text{pw}}$ .

# 3 Discussion

- 10 When using PDFs approach I, each data point is weighted with equal weight. This is different from approaches in which regression functions are applied. For example, in linear regressions which would be applied for the state independent case, the frequently applied regression method of ordinary least squares (OLS) minimises the sum of squared residuals. For the pre-conditioned case (regression lines in the ΔT<sub>g</sub> ΔR<sub>[X]</sub>-space have to pass through the origin, thus a = 0) only the slope of the regression line is determined, that is also equivalent to S<sub>[X]</sub>. We briefly illustrate the fact that in such linear regressions 15 data points further away from the origin get a higher weight by having a closer look on the most simplistic case with only 2
- data points, i = 1, 2 with point  $i: \Delta T_i, \Delta R_i, S_i = \Delta T_i / \Delta R_i$  OLS calculates a mean slope or S after

$$S = \frac{\Delta T_1 \cdot \Delta R_1 + \Delta T_2 \cdot \Delta R_2}{(\Delta R_1)^2 + (\Delta R_2)^2}$$
(12)  
=  $\frac{\Delta S_1 \cdot (\Delta R_1)^2 + \Delta S_2 \cdot (\Delta R_2)^2}{(\Delta R_1)^2 + (\Delta R_2)^2},$ (13)

from which one can easily determine that the squares in the equation lead to a higher weight for data points further away from 20 the origin.

One interesting effect of the application of regression statistics (being either linear, or non-linear) is the fact that the uncertainties in the parameters values to this higher order polynomial are very small, compared to the more general uncertainties (width of the PDF) in approach I.

#### 4 Conclusions

- 25 We find the following conclusions:
  - 1. In case no state dependency in  $S_{[X]}$  is found the data in the  $\Delta T_g \Delta R_{[X]}$  can be analysed by linear regressions to derive the slope that is after  $S_{[X]} = \frac{\Delta T_g}{\Delta R_{[X]}}$  an estimate of  $S_{[X]}$ . Be aware that approaches that combine point-wise-derived values of  $S_{[X]}$  in a probability density function to a more general number treat all individual data points with the same weight, while data far away from the origin get a higher weight in linear regression analysis using OLS.

- - 2. If state dependency of (specific) equilibrium climate sensitivity is found one has to be careful when quantifying equilibrium climate sensitivity  $S_{[X]}$ . The results based on local slopes are not directly comparable to estimates based on models or point-wise calculations, they can however be transferred into each other: the results based on local slopes transfer in results based on models or on the point-wise analysis by calculating the integral over the radiation perturbation interval following Eq. 9.
- 5

- 3. Taken at face value, the more sophisticated quantification of  $S_{[CO_2,LI]}$  as a function of radiative forcing perturbation  $\Delta R_{[CO_2,LI]}$  obtained here from the data sets described in Köhler et al. (2015) leads to numbers, which are by a factor of about 2 to 2.7 higher during climate conditions representing interglacials of the Pleistocene (last 2.1 Myr) than during full glacials during this period (Fig. 4).
- 4. A previous approach based on first-order local slopes (von der Heydt et al., 2014) already suggested higher S<sub>[CO2,LI]</sub> during interglacials than during glacials for data of the last 800 kyr, though only by 40%, while other approaches (Martínez-Botí et al., 2015) did not considered state dependency within the data set covering the last 800 kyr. These previous results agree more with the results we obtain for full glacial conditions. This might be the case, because linear regressions might not be forced through the origin (parameter *a* was not necessarily zero). When comparing results based on PDFs with the other approaches (as done in Figure 9 in Köhler et al. (2015)) the difference might also be explained because in linear regressions data points further away from the origin get a higher weight. The reason of this cold bias is not necessarily caused by the fact that most data are available for cold climates, as data may be binned (von der Heydt et al., 2014; Köhler et al., 2015).
  - 5. Interglacial climate conditions ( $\Delta R = 0 \text{ W m}^{-2}$ , Fig. 4) have a specific equilibrium climate sensitivity  $S_{[CO_2, LI]}$  between 2.0 K W<sup>-1</sup> m<sup>2</sup> (data of the last 2.1 Myr based on Hönisch-lab CO<sub>2</sub>-proxies) and 2.7 K W<sup>-1</sup> m<sup>2</sup> (ice core CO<sub>2</sub> data of the last 800 kyr). They are clearly at the upper end of ranges of  $S_{[CO_2, LI]}$  reported so far from various data sets of the last 65 Myr, e.g. in PALAEOSENS-Project Members (2012), Figure 3, the 95% probability in  $S_{[CO_2, LI]}$  ranged from 0.48 to 1.91 K W<sup>-1</sup> m<sup>2</sup>. Transfering these results based on data of the last Pleistocene to the near future remains difficult, since these palaeodata cover mainly conditions with negative radiative forcing anomaly, while for the future positive radiative forcing anomalies related to a rise in CO<sub>2</sub> are of interest, which obstructs a comparison as we showed that  $S_{[CO_2, LI]}$  depends on  $\Delta R_{[CO_2, LI]}$ . Furthermore,  $S_{[CO_2, LI]}$  needs to be corrected by factors related to fast and slow feedbacks to derive  $S^a$ , the actual or Charney climate sensitivity that is also constrained with climate models (see Equation 2 in PALAEOSENS-Project Members, 2012). However, such corrections and a detailed calculation of  $S^a$  are not readily available for  $\Delta R_{[CO_2, LI]} = 0 \text{ W m}^{-2}$  and a detailed calculation of  $S^a$  is beyond the scope of this technical note. Nevertheless does the palaeodata-based analysis suggest that the equilibrium climate sensitivity for present-day based on palaeodata is more at the high end with respect to reported values in the IPCC AR5 report (e.g. Thematic Focus Element 6 in Stocker et al., 2013).

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
