# Peer review of "Technical Note: Calculating state dependent equilibrium climate sensitivity from palaeodata"

_Climate of the Past, 2016_

## Referee Comment (RC1) · Anonymous Referee #1 · 19 Apr 2016

Review of "Calculating state dependent equilibrium climate sensitivity from palaeodata" by Köhler et al.

This paper compares methods for evaluating equilibrium climate sensitivity using as an example the datasets of radiative forcing and global mean surface temperature change of Köhler et al (2015) for climate states over the last 800 kyr. If I have understood correctly, the first method is to calculate the ratio of temperature change to forcing change, both evaluated with respect to a reference state (or regress one against the other, requiring zero intercept), and the second method is to calculate the derivative of temperature change with respect to forcing change, without need of a reference state. They refer to the quantity estimated i.e. the global surface warming per unit increase in forcing as the "specific equilibrium climate sensitivity", whereas in the literature relating to climate projections e.g. in the IPCC reports, this quantity is called the "climate

sensitivity parameter".

If the climate sensitivity parameter is a function of climate state, the methods give different results. One could argue that they give different quantities, though as the authors point out they can be related by integrating along a trajectory. I don't think it is clear that one or the other should be preferred. It depends on the purpose. It is important to be aware of this, of course, when using palaeo-data to constrain future projections, as the authors suggest in their conclusions.

I note that all the palaeoclimate states are assumed to be equilibria for the atmosphere-ocean system. That, they assume there is no heat storage occurring in the ocean. If there is, it has to be subtracted from the forcing in order to estimate the climate sensitivity parameter. AOGCMs suggest that the ocean takes more than 1000 years to reach a steady state after radiative forcing is changed, with everything else held constant. It may be worth discussing this point.

An analogous question of whether the climate sensitivity is defined by the slope from the origin to the endpoint, or by the tangent slope, arises in consideration of AOGCM simulations, for example under constant 4xCO2, as they approach equilibrium e.g. Gregory et al. (2004, 10.1029/2003gl018747), Li et al. 2012 (10.1007/s00382-012-1350-z). In most AOGCMs the slope is found not to be constant, and is a function of state or time e.g. Andrews et al. (2012, 10.1029/2012GL051607). The reasons are probably not the same as on the multimillennial timescale, but the technical issue is similar.

I think the technical point of the paper is sound, but I would say that it seems rather laboured, and I feel it could be written more briefly. The discussion section 3 could be incorporated in 2.1 if it refers only to the first method. However one could also remark that regression might be used to determine local slopes in 2.2, and the same issue applies that the extremes get more weight.

The technical issue outlined in the abstract is summarised by points 1 and 2 of the conclusions. The majority of the conclusions are about implications for the interpretation of palaeoclimate sensitivity and particularly about the dataset of Köhler et al. While these points may be fine, it seems to me that they are not really conclusions of the technical discussion. They are more of a discussion of the scientific implications for the particular case considered.

Point 5 in particular raises more subjects. Should one expect the climate sensitivity parameter evaluated from the palaeorecord to be applicable to the future? How should account be taken of forcings apart from CO2 and ice-sheet albedo? There is a lot of other literature about the dependence of the climate sensitivity or feedback parameter on the nature of the forcing agent, and about its dependence on climate state. The authors mention the need to remove "slow" feedbacks to make their evaluation comparable with AOGCM evaluations, but this is inconsistent with their regarding ice-sheets as a forcing (rather than as a slow feedback), I would say. Is their quantity a climate sensitivity or an Earth system sensitivity? These are important questions, but not the stated subject of this technical note, and they do not appear in the abstract. I feel therefore that either the discussion should be restricted to the technical point, or that the scope of the paper as represented by the title and the abstract should be widened, and a fuller discussion of the implications should be included before the conclusions are reached.

Minor comments

p1 line 10. "One prominent approach". Please could references be given.

p2 line 7. "Results point more and more in the direction". Again, please could references be given.

p4 line 12. I don't think radiative forcing is ever "absolute"; it is always referred to some climate state e.g. in IPCC reports to pre-industrial (c 1750).

p5 line 4. I think it should be "mean local slope".

---

## Referee Comment (RC2) · J. Bloch-Johnson (Referee) · 27 Apr 2016

Review of "Technical Note: Calculating state dependent equilibrium climate sensitivity from palaeodata", Köhler et al.

The authors, in a recent paper (Köhler et al., 2015), found evidence for state dependence of climate sensitivty in the paleoclimate record. In the present work, the authors explore an important consequence of this state-dependence, namely that point-wise and slope-based definitions of climate sensitivity can differ. This technical issue can present complications, and the authors discuss potential outcomes of these complications, along with ways of reconciling estimates made using one of these definitions with the other. The authors are performing needed and useful work by raising and addressing these complications, as the answers to the questions that this paper aims to present would be of use the broader community looking to study climate sensitivity and its state dependence. Unfortunately, I believe that there is a fundamental problem in how the authors approach their study of this state dependence.

Measuring how the climate sensitivity changes with state requires a formal definition of what we mean by "state-dependence". Although climate state can refer to a number aspects of the climate, it is principally taken to refer to the surface temperature. For example:

- "Climate sensitivity may also be time-dependent and state-dependent; for example, in a much warmer with little snow and ice, the surface albedo feedback would be different from todays." (Knutti and Hegerl, 2008)

- "Consistent intercomparison is crucial to detect systematic differences in sensitivity values — for example, due to changing continental configurations, different climate background states, and the types of radiative perturbations considered. These differences may then be evaluated in terms of additional controls on climate sensitivity, such as those arising from plate tectonics, weathering cycles, changes in ocean circulation, non-$CO_2$ greenhouse gases (GHGs), enhanced water-vapour and cloud feedbacks under warm climate states. Palaeoclimate data allow such investigations across geological episodes with very different climates, both warmer and colder than today."; "...only ancient records provide insight into climate states globally warmer than the present."; "...it may be more useful to consider past warm climate states as test-beds for evaluating processes and responses.." (PALAEOSENS, 2012)

- "The available evidence suggests that climate sensitivity depends considerably on the reference climate state... In particular, estimates of the ECS from climates much warmer than today, such as the Paleocene-Eocene, would naturally yield higher values..."; (Meraner et al., 2013)

- "The climate sensitivity decreases as the system warms after forcing is stabilized at S2050 and S2100 levels." ("Climate sensitivity and climate state," Boer and Yu (2003))

There is a physical reason for giving pride of place to surface temperature, as suggested by the above statements: the feedbacks that determine the climate sensitivity, such as the water vapor, surface albedo, and cloud feedbacks are expected to change with the surface temperature, and so will make climate sensitivity state-dependent. As a result, many papers

make the assumption the most direct way to capture the state-dependence of climate sensitivity in a simple model is to add a polynomial dependence of the climate feedback parameter on temperature (e.g., Roe (2009), Colman and McAvaney (2009), Jonko et al. (2012)).

Rather than taking this course, the present authors test the polynomial dependence of climate sensitivity on radiative forcing. While radiative forcing and surface temperature change are expected to change proportionally for small forcings, for large forcings this proportionality breaks down, and during bifurcations the relationship between the two can become poorly defined, or at least discontinuous. As a result, for these larger forcings, analyses of state-dependence that measure dependence on radiative forcing rather than temperature can give quite different answers than the temperature-dependent approach. They can less faithfully capture the effects of changes in feedback strength, and generally cannot represent the effects of bifurcations. Worse still, they can give unphysical answers. I will discuss examples of these three issues below, but first I will more formally describe the temperature-dependence approach taken by previous authors.

**1 The temperature-dependent approach**

The climate sensitivity formalism is built around the idea that Earth can be modelled by a single ordinary different equation, $cdT/dt = N$, where $N$ is the net top-of-atmosphere radiative flux, $c$ is some thermal inertia of the Earth and $T$ is the surface temperature. The value of $N$ depends on various forcing agents such as the $CO_2$ concentration (or, depending on the partition between feedbacks and forcing, land ice), which we will denote as $F$, as well as on aspects of the general state of the atmosphere and surface that are expected to change with the surface temperature, $T$. In other words, $N(T, F)$, and for a given forcing, $N$ is simply a function of $T$. This latter dependence has been an essential assumption from the earliest energy balance models (e.g., North (1975): "In order to set up a simple energy-balance climate model it is necessary to assume that all energetic fluxes can be parameterized by the temperature at the earth's surface.")

For a preindustrial state $T_{pre}$ and $F_{pre}$, $N(T_{pre}, F_{pre}) = 0$. For small enough changes, the cross terms between $T_{pre}$ and $F_{pre}$ will be small[1], and we will have $N(T, F) = N(T, F_{pre}) + \Delta R$, where $\Delta R$, the radiative forcing, can be defined to be $N(T_{pre}, F) - N(T_{pre}, F_{pre})$. If we impose a positive radiative forcing, it makes $N$ positive, causing the Earth to warm. $\partial N/\partial T|_{pre}$ typically has a negative slope, so that the resulting warming compensates for the radiative forcing until $N(\Delta T + T_{pre}, \Delta R + F_{pre}) = 0$ and the system is in equilibrium.
* * *
[1]These cross terms can be thought of as representing how changes to the forcing elements effect the ability of the surface to set the top-of-atmosphere energy flux. For example, for a planet with a richer $CO_2$ atmosphere, the same change in surface temperature might have less of a direct Planck effect on the outgoing radiation.

In other words, the climate sensitivity (the change in surface temperature $\Delta T$ due to a given radiative forcing $\Delta R$) is determined by the function $N(T, F_{pre})$ — the dependence of top-of-atmosphere fluxes on surface temperature. As described in Lorius et al. (1990), "The response of the system to an increase of radiative forcing would be a change in [equilibrium surface temperature] $T_e$ necessary to restore radiative equilibrium." We typical linearize this dependency, taking $N(\Delta T + T_{pre}, \Delta R + F_{pre}) = \lambda \Delta T + \Delta R$ where $\lambda = \partial N/\partial T$, so that the climate sensitivity $S = -1/\lambda$. However, the climate sensitivity might change with climate state, because the dependence of top-of-atmosphere flux on surface temperature changes with climate state. (This is what is meant by the change in feedbacks discussed above). The most natural way to capture this change is to take the next term in the Taylor expansion of $N(T, F_{pre})$, namely $N(\Delta T + T_{pre}, \Delta R + F_{pre}) = \lambda \Delta T + a\Delta T^2 + \Delta R$ (e.g., supplementary materials for Roe and Baker (2007), Roe (2009), Colman and McAvaney (2009), Jonko et al. (2012)). This Taylor expansion could potentially be expanded further.

**2    Advantages of fitting temperature-dependent approach over the state-dependent approach**

For these reasons, even though it may seem natural to think of $\Delta T$ as a function of $\Delta R$ and to take a polynomial fit of $\Delta T$ to $\Delta R$, we do not expect the relationship $\Delta T(\Delta R)$ to be polynomial, or even necessarily a function. As a result, diagnosing state-depenence of climate sensitivity with a polynomial fit of $\Delta T(\Delta R)$ can lead to a number of problems:

**2.1    $\Delta T(\Delta R)$ is not necessarily a well-defined function**

If there are any sorts of bifurcations in the Earth's climate — and given the Snowball Earth and the runaway greenhouse effect, we expect there are — $\Delta T(\Delta R)$ will not be a well-defined function, since for these cases the same radiative forcing can lead to very different levels of warming depending on the path taken by that forcing. Bifurcations need not be large events like Snowball Earth or the runaway greenhouse; a example of a more modest jump in temperature is given in previous work by some of the authors, Heydt et al. (2014), which uses a simple four-box model of the Earth (Gildor and Tziperman (2001)) to explore the behavior of glacial/interglacial cycles. I will illustrate my point about $\Delta T(\Delta R)$ by using this jump as an example.

Consider Figure 1, which uses Figure 1f from Heydt et al. (2014) (reproduced here as 1a) to estimate $N(T, F)$ for that model (given in 1b):

[Figure]

Figure 1: The presence of bistability can allow the same quantity of radiative forcing to cause different changes in surface temperature depending on how the forcing is administered. In an example drawn from Heydt et al. (2014), a direct forcing of $-3W/m^2$ from preindustrial causes $1.3K$ of cooling (panel c), while a direct forcing of $-4W/m^2$ (panel d) followed by a subsequent forcing of $1W/m^2$ (panel e) causes a total $2.9K$ of cooling.

If we start in equilibrium at preindustrial conditions, apply a radiative forcing of $-3W/m^2$, and let the system cool until it reaches equilibrium, we get a cooling of $1.3K$ (1c). If we instead first apply a radiative forcing of $-4W/m^2$ and let the system equilibrate (1d), and then apply a radiative forcing of $1W/m^2$ and the let the system reequilibrate, the system will have cooled by $2.9K$, even though the overall forcing is still $-3W/m^2$.

This circumstance cannot be captured by a polynomial fit of $\Delta T(\Delta R)$, or by any sort of function $\Delta T(\Delta R)$ unless we assume that forcing can only move in one direction, which is inappropriate for glacial/interglacial cycles. On the other hand, a polynomial fit to $N(T, F)$ could capture this behavior easily (e.g. with a cubic function).

**2.2 Different values of $\Delta R$ are very unlikely to be able to cause the same $\Delta T$**

Some of the polynomial fits in the present paper (the red curve in Figure 1 and Figure 3a, and the black curve in Figure 3b) and in Köhler et al. (2015) (Figure 7a, some of the curves in 7e) suggest that different values of radiative forcing $\Delta R$ can cause the same temperature change $\Delta T$. I would argue this is almost certainly not a possibility.

Suppose there were two radiative forcings $\Delta R_1$ and $\Delta R_2$ that both cause the same $\Delta T$, where $\Delta R_1$ is less than $\Delta R_2$. Suppose we first impose the $\Delta R_1$ forcing, causing the planet to come to equilibrium after a temperature change of $\Delta T$. Since we are in equilibrium, $N$ is now $0$ again. Now suppose that we impose a further $\Delta R_2 - \Delta R_1$ forcing. Since $N$ is now $\Delta R_2 - \Delta R_1 > 0$, the Earth will warm. However, it has to eventually cool again to return to the same temperature it had before the $\Delta R_2 - \Delta R_1$ was imposed in order for $\Delta R_2$ to also cause a total warming of $\Delta T$. In order for this to happen, N has to go from being positive to negative at some point, in order for the planet to start cooling. However, this requires that at some point $N = 0$. As soon as it does, the Earth would be in equilibrium at a warmer temperature than $\Delta T$, leading to a contradiction. Only when the overall zero-dimensional energy balance formalism breaks down (e.g. through having very different spatial temperature patterns with quite different $N(T, F_{pre})$ for the same value of $T$) could this contradiction be circumvented, an intriguing possibility but not one that I have seen much evidence for.

As a result, these polynomial fits are almost certainly unphysical. It should be noted that for Köhler et al. (2015), this unphysical behavior occurs not in regions of the fit that are extrapolations or have a small amount of data, but in the heart of the regions for which we have data.

**2.3 Using temperature-dependence results in more plausible and coherent fits**

To illustrate the practical significance of using temperature rather forcing-dependent estimates of sensitivity, I've recreated Figure 7 from Köhler et al. (2015), with polynomial fits performed using both methods. The data was obtained using the relative spatial position of the SVG elements from Figure 7 and then fitting them to the given values in the graph. As a result, the data is not necessarily precisely the same as that used in Köhler et al. (2015), but is good enough for our illustrative purposes. For the forcing-dependent fits (black lines), I've used the degree of the polynomial used in Köhler et al. (2015). For the temperature-dependent fits (blue lines), I fit $\Delta R = -N(T, F_{pre}) = -(\lambda \Delta T + a \Delta T^2 + b \Delta T^3)$:

[Figure]

Figure 2: A reconstruction of Figure 7 from Köhler et al. (2015) (see text). The four columns correspond to four paleoclimate datasets. In the top row, land ice changes are considered a feedback on the climate system, while in the bottom row it is considered a forcing. Black lines show reconstructions of the forcing-dependent fits from Köhler et al. (2015), while blue lines show cubic fits to $\Delta R = -N(T, F_{pre}) = -(\lambda \Delta T + a \Delta T^2 + b \Delta T^3)$.

The fits both capture roughly the same amount of variance of their respective dependent variables ($\Delta R$ for the temperature-dependent fits, $\Delta T$ for the forcing-dependent fits) (though it should be noted that minimizing errors in $\Delta R$, as is done by the temperature-dependent fit, can be much more significant than minimizing errors in $\Delta T$, since $S = \Delta T / \Delta R$). Further, for the Hönlisch and ice core datasets (which are also used in the present article), there is not much difference in the nature of the fits (except for change in the sign of the slope of the $\Delta T(\Delta R)$ slope in the upper ice cores figure, which I argued above was unphysical).

However, the two Foster analyses and the bottom Pagani analysis give quite different answers depending on what method you use. Note that these methods do show improvements in $R^2$, and do appear visually to better capture the shape of the underlying data. However, there's an even more significant result of this change: for the temperature-dependent approach, **all the datasets which can be fit (all except the upper Pagani figure) agree that the Earth has a lower sensitivity for colder periods**, and in the case where the land ice changes are treated as a forcing rather than a feedback, the sensitivity is significantly smaller. This is seen in Figure 3, which compares the climate sensitivities ($d\Delta T/d\Delta R$) and climate feedback parameters ($d\delta R/d\Delta T$) for the four datasets (different colors) and different approachs (forcing-dependent approach, dashed lines; temperature-dependent approach, solid

line). The temperature-dependent approach, with the exception of the top Pagani data, gives a similar estimate of sensitivity and its state-dependence for some of the temperature range, while the forcing-dependent fits disagree about the size of the sensitivity and how it changes in a colder/less-forced world.

[Figure]

Figure 3: Climate sensitivities ($d\Delta T/d\Delta R$) and climate feedback parameters ($d\delta R/d\Delta T$) for the four datasets (colors from Figure 2) and different approachs (forcing-dependent approach, dashed lines; temperature-dependent approach, solid lines). Aside from the $\Delta R_{[CO_2]}$ Pagani case, for which the fit itself is quite poor, the different temperature-dependent fits roughly agree on the size and change of climate sensitivity and feedback parameter under different temperatures, in contrast with the different forcing-dependent fits.

Köhler et al. (2015) find that the Foster and Pagani datasets disagree with the ice cores and Hönisch datasets as to whether climate sensitivity is state-dependent, and as to the manner of this state-dependence. The temperature-dependent approach instead suggests that these datasets all agree: the climate sensitivity is smaller in climate states colder than present.

**3 Conclusions**

The authors raise important points about the need to reconcile point-wise and slope-wise estimates of climate sensitivity when that sensitivity is state-dependent. However, I have strong concerns about the fundamental approach taken by the authors with respect to formalizing this phenomenon. The points I raise in my response may seem pedantic, particularly given that for the fits they have included in the present work, the temperature-dependent and forcing-dependent approaches give very similar answers. However, as the examples above show, the distinction is important: the forcing-dependent approach can generate unphysical estimates of the Earth's behavior, and miss important nonlinear behavior. I would argue that their earlier work could have found a coherence between datasets that was missed.

I would either hope that the authors would rethink their use of forcing-dependence, or would justify their approach in a manner that answers the concerns raised in this response. I have a couple of other remarks about the content of this paper: A case study showing the use of the sorts of equations found in Section 2.3 would be useful. It would also be useful to see some of the points saved for the conclusions (4 and 5 in particular) discussed more fully in the body of the paper.

Once more, I feel that this paper is a needed one that builds on important work produced previously by the authors. I look forward to seeing how it develops.

Sincerely, Jonah Bloch-Johnson

**References**

Boer, G. J. and B. Yu (2003). "Climate sensitivity and climate state". en. In: *Climate Dynamics* 21.2, pp. 167–176.

Colman, R. and B. McAvaney (2009). "Climate feedbacks under a very broad range of forcing". en. In: *Geophysical Research Letters* 36.1, p. L01702.

Gildor, H. and E. Tziperman (2001). "A sea ice climate switch mechanism for the 100-kyr glacial cycles". en. In: *Journal of Geophysical Research: Oceans* 106.C5, pp. 9117–9133.

Heydt, A. S. von der et al. (2014). "On the state dependency of fast feedback processes in (paleo) climate sensitivity". en. In: *Geophysical Research Letters* 41.18, pp. 6484–6492.

Jonko, A. K. et al. (2012). "Climate Feedbacks in CCSM3 under Changing CO2 Forcing. Part I: Adapting the Linear Radiative Kernel Technique to Feedback Calculations for a Broad Range of Forcings". In: *Journal of Climate* 25.15, pp. 5260–5272.

Knutti, R. and G. C. Hegerl (2008). "The equilibrium sensitivity of the Earth's temperature to radiation changes". en. In: *Nature Geoscience* 1.11, pp. 735–743.

Köhler, P. et al. (2015). "On the state dependency of the equilibrium climate sensitivity during the last 5 million years". In: *Clim. Past* 11.12, pp. 1801–1823.

Lorius, C. et al. (1990). "The ice-core record: climate sensitivity and future greenhouse warming". en. In: *Nature* 347.6289, pp. 139–145.

Meraner, K., T. Mauritsen, and A. Voigt (2013). "Robust increase in equilibrium climate sensitivity under global warming". en. In: *Geophysical Research Letters* 40.22, pp. 5944–5948.

North, G. R. (1975). "Theory of Energy-Balance Climate Models". In: *Journal of the Atmospheric Sciences* 32.11, pp. 2033–2043.

PALAEOSENS (2012). "Making sense of palaeoclimate sensitivity". en. In: *Nature* 491.7426, pp. 683–691.

Roe, G. (2009). "Feedbacks, Timescales, and Seeing Red". In: *Annual Review of Earth and Planetary Sciences* 37.1, pp. 93–115.

Roe, G. H. and M. B. Baker (2007). "Why Is Climate Sensitivity So Unpredictable?" en. In: *Science* 318.5850, pp. 629–632.

---

## Referee Comment (RC3) · Anonymous Referee #2 · 28 Apr 2016

I just wanted to add a postscript: I responded with a similar point to the previous paper submitted by this group (Köhler et al, 2015), and in their response the authors suggested that the approach I was discussing was valid for transient but not equilibrium climate sensitivity. I wanted to be explicit that the concerns I have raised now and then were all concerned with equilibrium climate sensitivity, not transient sensitivity. Curves of N(T,F) contained in this response (e.g. Figure 1) are not meant to represent fits to Gregory plots (Gregory et al, 2004), but rather represent the value of top-of-atmosphere energy flux a world in equilibrium at temperature T would have if you abruptly changed the forcing agents to be F. In that sense, they determine the equilibrium response to a forcing, and plots of time series of T vs. N would not necessarily follow these curves.

Cheers, Jonah Bloch-Johnson

---

## Referee Comment (RC4) · Anonymous Referee #3 · 2 May 2016

Note: I have not read any of the other comments or reviews in the interactive review before posting this review.

This paper appears to be a comment on, or clarification of, the methods used in Kohler et al 2015. It is not clearly written and this has made the reviewing a bit tricky. I had to go back and read Kohler et al 2015 to have a clue what this is about. It transpires that Kohler et al 2015 and von de Heydt et al 2014 use different methods to calculate the climate sensitivity, and I think the point of this manuscript could be to highlight the differences in the results obtained by using the different methods. But I'm not quite sure. It is stated in the manuscript that von de Heydt et al used Approach II, but the authors do not clearly state that Kohler et al used Approach I. Although this manuscript is underwhelming, I think it may help avoid future problems and confusions by explicitly pointing out the differences in the two methods. It is a shame that this work was not

included as an appendix to Kohler et al 2015!

So of course I had to go back to Kohler et al 2015. In that paper, curves are fitted to temperature / radiative forcing (hereafter RT) plots. They conclude that, in some cases, nonlinear fits may be appropriate and thus climate sensitivity is state dependent. There seem to be two problems with this conclusion. The first is the statement in the introduction of Kohler et al 2015, "However, we are not aware that a difference in the response has been shown for radiative forcing from surface albedo changes ( R[LI]) and $CO_2$ ( R[CO2] ). Hence we combine them linearly." A different response to these two forcings was already clearly shown in Yoshimori et al (doi:10.1175/2011JCLI3954.1, 2011). Thus, since the RT curves for "LI" and "CO2" are likely to be different, finding that the RT curves when CO2 and LI forcings are both included is nonlinear does not uniquely show state dependence of climate sensitivity. Rather it more likely shows a combination of state dependence and forcing dependence. The second potential problem in Kohler et al 2015 that is pertinent to the manuscript under review is possible over-fitting of high order polynomials. More parameters means it is easier to fit a scatter of points, and the method used to discriminate between the polynomials should take this into account. I'm not sure exactly how the authors employed the F-test, but why did they not use something like BIC (Bayesian Information Criterion) which explicitly takes into account this over-fitting problem? It seems likely that the higher order polynomials are not supported by the data. Since the authors are in the mode of commenting on their own previous work, perhaps they could also address this over-fitting issue in the manuscript under review.

The main thrust of the manuscript under review is that, in the case of a nonlinear polynomial in RT space, a different result for calculation of the gradient of a curve will be obtained depending on which position on the polynomial you start from. This is obvious. However, the point that I think the authors are making is that Kohler et al 2015 calculated all their values of R relative to a particular state, (Ro,To) and then calculate Sensitivity (S) as (T-To)/(R-Ro) whereas Von de Heydt et al calculate the tangent to the

RT curve. These two methods result in a different function for S. The authors state that Approach I (Kohler et al 2015) is the most robust approach, but it is not clear why, and it seems to me that representing S as dT/dR (Approach II of von der Heydt et al) is more generally our scientific goal. The authors state that Approach II "is more readily comparable to climate model results". This is an odd statement as it really depends on the climate model experiment. It would be possible to exactly reconstruct Approach I using a climate model, and indeed a common suite of experiments (0.5CO2, 2xCO2, 4xCO2 starting from try control state), are a version of Approach I. I don't understand the point of the Discussion section. It isn't a discussion of the previous sections, but another comment on a different part of the analysis. It seems to be comparing two completely different things. One thing is the fitting of a curve to the scatter of points in RT space, which results in the calculation of a functional relationship between S and T. The second thing is taking the scatter of points in the RT space and turning them into a distribution in R (or S) space, which indicates how often the earth system has wandered into different parts of RT space in the paleoclimate record.

---

## Author Comment (AC1) · 18 May 2016

**Response to Reviewer #1**

1.1 This paper compares methods for evaluating equilibrium climate sensitivity using as an example the datasets of radiative forcing and global mean surface temperature change of Kohler et al (2015) for climate states over the last 800 kyr. If I have understood correctly, the first method is to calculate the ratio of temperature change to forcing change, both evaluated with respect to a reference state (or regress one against the other, requiring zero intercept), and the second method is to calculate the derivative of temperature change with respect to forcing change, without need of a reference state. They refer to the quantity estimated i.e. the global surface warming per unit increase in forcing as the "specific equilibrium climate sensitivity", whereas in the literature relating to climate projections e.g. in the IPCC reports, this quantity is called the "climate sensitivity parameter".

**Our reply:** Unfortunately, the definition of variables is not always the same. This is especially the case for the global surface warming ($\Delta$T) per unit increase in forcing ($\Delta$R) which we call "specific equilibrium climate sensitivity" and which we give the notation $S$. The reviewer correctly noted that in the IPCC reports, this quantity is called the "climate sensitivity parameter". However, it is also given in the IPCC reports the acroynm $\lambda = \lambda_1 = \Delta T/\Delta R$. Other studies, e.g. *Dufresne and Bony* (2008) call $\lambda$ the "climate feedback parameter", $\lambda = \lambda_2 = -\Delta R/\Delta T$. As can be seen $\lambda_1 = -1/\lambda_2$. To avoid confusion we wanted to get away from the notation $\lambda$ and decided to stick the nomenclature brought up by the review of *PALAEOSENS-Project Members* (2012), in which $S_{[X]}$ was introduced, with X being the process(es) whose forcing is explicitly considered in the calculation. Furthermore, since $S_{[X]}$ in the context of complex climate models used for IPCC is not a simple (tunable) *parameter* of these models, but an outcome of the analysis of simulations, we find the wording "climate sensitivity parameter" confusing. Nevertheless, we will briefly mention now in the introduction the different wording used within the IPCC.

1.2 If the climate sensitivity parameter is a function of climate state, the methods give different results. One could argue that they give different quantities, though as the authors point out they can be related by integrating along a trajectory. I don't think it is clear that one or the other should be preferred. It depends on the purpose. It is important to be aware of this, of course, when using palaeo-data to constrain future projections, as the authors suggest in their conclusions.

**Our reply:** This is exactly our main motivation of writing this technical note: We need to be aware of the different approaches and how they might be transfered into each other. We will revise the draft for clarity and that per se no approach might be preferred.

1.3 I note that all the palaeoclimate states are assumed to be equilibria for the atmosphere-ocean system. That, they assume there is no heat storage occurring in the ocean. If there is, it has to be subtracted from the forcing in order to estimate the climate sensitivity parameter. AOGCMs suggest that the ocean takes more than 1000 years to reach a steady state after radiative forcing is changed, with everything else held constant. It may be worth discussing this point.

**Our reply:** Transient effects were investigated and the equilibrium assumption was checked in previous studies. In a previous analysis to which some of the authors contributed (*PALAEOSENS-Project Members*, 2012) we investigated how important the contribution of data-points from periods of abrupt climate change are for the calculation of $S_{[X]}$. For that aim we filtered time series of the last 800,000 years for periods in which global temperature change was fast, e.g. faster than 0.5 or 0.1 K per 1 kyr. This would filter out millenium-scale variability, so-called Dansgaard-Oeschger event, for which we would expect that climate was indeed not in equilibrium, as suggested by this comment. However,

we concluded in *PALAEOSENS-Project Members* (2012) that within the given data sets transient effects are not important and the assumption of equilibrium is valid. Here, we investigate in detail different time series than in this previous study. However, this assumption of equilibrium conditions seems still valid, also because a lot of the time series investigated here ($\Delta T_g$, $\Delta R_{[LI]}$) are based on the model-based inversion of a benthic $\delta^{18}$O stack with 3-D ice sheet models. Within such a setup, from which we took output at discrete time steps of 2,000 years, abrupt climate change connected with Dansgaard-Oeschger events are per se not included. This will now briefly be discussed just before subsection 2.1 starts.

1.4 An analogous question of whether the climate sensitivity is defined by the slope from the origin to the endpoint, or by the tangent slope, arises in consideration of AOGCM simulations, for example under constant 4xCO2, as they approach equilibrium e.g. Gregory et al. (2004, 10.1029/2003gl018747), Li et al. 2012 (10.1007/s00382-012- 350-z). In most AOGCMs the slope is found not to be constant, and is a function of state or time e.g. Andrews et al. (2012, 10.1029/2012GL051607). The reasons are probably not the same as on the multimillennial timescale, but the technical issue is similar.

**Our reply:** Thanks for these suggestions. We will briefly mention these simulation results in the revised paper.

1.5 I think the technical point of the paper is sound, but I would say that it seems rather laboured, and I feel it could be written more briefly. The discussion section 3 could be incorporated in 2.1 if it refers only to the first method. However one could also remark that regression might be used to determine local slopes in 2.2, and the same issue applies that the extremes get more weight.

**Our reply:** We will follow this comment (a similar comment was made by reviewer 3) and we will delete the discussion section. Details so far mentioned there will be either shifted to places where issues are first mentioned or they will be deleted. However, please note that there are no guidelines yet how long a technical note should be. We believe that a lot more shortening will be difficult, since this might results in a piece of work that no longer stands on its own.

1.6 The technical issue outlined in the abstract is summarised by points 1 and 2 of the conclusions. The majority of the conclusions are about implications for the interpretation of palaeoclimate sensitivity and particularly about the dataset of KoÌĹhler et al. While these points may be fine, it seems to me that they are not really conclusions of the technical discussion. They are more of a discussion of the scientific implications for the particular case considered.

**Our reply:** The reviewer is correct that some of the points in the conclusion section are no true conclusions of this short technical note, but more a discussion of potential implications. However, we still believe these points should be briefly mentioned, because only after applying these different methods to some data one really see the consequences / implications. Therefore, we decided to revise the final section into something called *Discussions, Conclusions, Implications*. The abstract will now also include a sentence saying that we indicate some implications if the approaches are applied to Pleistocene data.

1.7 Point 5 in particular raises more subjects. Should one expect the climate sensitivity parameter evaluated from the palaeorecord to be applicable to the future? How should account be taken of forcings apart from CO2 and ice-sheet albedo? There is a lot of other literature about the dependence of the climate sensitivity or feedback parameter on the nature of the forcing agent, and about its dependence

on climate state. The authors mention the need to remove "slow" feedbacks to make their evaluation comparable with AOGCM evaluations, but this is inconsistent with their regarding ice-sheets as a forcing (rather than as a slow feedback), I would say. Is their quantity a climate sensitivity or an Earth system sensitivity? These are important questions, but not the stated subject of this technical note, and they do not appear in the abstract. I feel therefore that either the discussion should be restricted to the technical point, or that the scope of the paper as represented by the title and the abstract should be widened, and a fuller discussion of the implications should be included before the conclusions are reached.

**Our reply:** This comment refers to the last point in our conclusion section. We have a different perception of what is mentioned (or not mentioned) in that paragraph than the reviewer. We actually think that this point only briefly summarizes what would be the next logical steps, one might want to go, but we do not follow on into any details, because we also think (and briefly write) that these further investigations are indeed very interesting and wanted, but not within the scope of such a short technical note. We therefore do not think that this point should be deleted, and it might fit into the final section now better since we will rephrase its title (see also our reply to the previous comment). We will therefore briefly mention now in the abstract some implications given here. However, we also see, that some clarifications are necessary, we therefore give some in-depth responses to every of the various subpoints.

*Should one expect the climate sensitivity parameter evaluated from the palaeorecord to be applicable to the future?*
One motivation for palaeo studies still is to learn from the past to better predict the future. As long as climate sensitivity can be assumed to be independent of climate state, this raises no problems. With the state dependency of climate sensitivity, however, the application from knowledge gained from past climates

to presumably different future climates is not straightforward anymore. We don't elaborate such a transfer here, but future studies will certainly investigate more details of this issue. Therefore, we believe our warning statements brought up here, that no easy transformation is possible is potentially still of interest for the reader. We also see this as a kind of final interpretation of what can be gained from our previous 2015 study. All that said, it probably would have been also a possible and elegant way to have all details of this technical note already included in this 2015 paper. However, we have to say that our analysis shown here was not that advanced and ready in 2015.

*How should account be taken of forcings apart from CO2 and ice-sheet albedo?* and *The authors mention the need to remove "slow" feedbacks to make their evaluation comparable with AOGCM evaluations, but this is inconsistent with their regarding ice-sheets as a forcing (rather than as a slow feedback), I would say.*

We think there exist some misunderstanding here. The concept of treating the land ice sheet albedo as a radiative forcing and not as a slow feedback is an operational decision which was widely explained in the *PALAEOSENS-Project Members* (2012). Also note, that this is already accounted for in our introduction in which we write that the land ice contribution is a correction of a slow feedback: *"Here, $X$ (of $S_{[X]}$) corresponds to the forcing processes considered, typically changes in (sometimes also including the other greenhouse gases and ), potentially corrected for some slow feedbacks such as planetary albedo changes causes by variations in land ice (LI), vegetation (VE) or dust (aerosols (AE))."* For more details on this issue please see the section **Forcing and slow feedbacks** in the PALAEOSENS paper (2012).

*% Is their quantity a climate sensitivity or an Earth system sensitivity?*

[Figure]

Interestingly, this question was already asked by the handling editor after submission and we repeat and extend our point of view on this issue:

The terms *Earth system sensitivity*, *Charney sensitivity* and even *climate sensitivity* estimate the mean global warming as a response to a doubling of $CO_2$ concentrations, so units of all these quantities are "$^o$C" or "K". We here follow the more general approach which in detail is here and elsewhere (e.g. PALAEOSENS paper in 2012 in Nature) calculating temperature change per radiative forcing change. Therefore, our result variable is in detail labelled *specific* equilibrium climate sensitivity $S$ (see beginning of abstract), which then is expressed in units "K W$^{-1}$ m$^2$", or as pointed out by the reviewer *climate sensitivity parameter* within the context climate simulation results used for the IPCC. Truely, Earth system sensitivity (or ESS) is a special case of $S$. To calculate ESS the specific equilibrium climate sensitivity $S$ is already multiplied by the radiative forcing caused by a doubling of atmospheric $CO_2$ concentration. In detail, the connection between both is established in the PALAEOSENS 2012 paper. We like to state our calculations in the more general formulation of the nomenclature put forward by the PALAEOSENS group in their Nature review paper, also because its definition is more strict and, if properly followed on, confusions are avoided more easily. Furthermore, since we never go that final step to calculate any temperature change out of $S$ because we believe that due to the non-linear character of the $S$ this is not straight-forward possible, we believe sticking to the nomenclature as used so far is correct and more general and should not lead to confusion. However the distinction to ESS will now briefly be mentioned in the introduction.

**1.8 Minor comments**

p1 line 10. "One prominent approach". Please could references be given.

**Our reply:** References given now (e.g. *Rohling et al.*, 2012; *von der Heydt et al.*,

2014; *Köhler et al.*, 2015; *Martínez-Botí et al.*, 2015).

p2 line 7. "Results point more and more in the direction". Again, please could references be given.

**Our reply:** References given now (e.g. *Crucifix*, 2006; *Hargreaves et al.*, 2007; Yoshimori et al., 2011)

p4 line 12. I don't think radiative forcing is ever "absolute"; it is always referred to some climate state e.g. in IPCC reports to pre-industrial (c 1750).

**Our reply:** In a climate model absolute radiative forcing can be calculated and we therefore think that our calculations in section 2.3, which rely on absolute radiative forcing should be included here. In principle it would be possible to reduce this section to calculations based on relative changes in radiative forcing only. However, such a shortening would unnecessarily miss some of the insights given here.

p5 line 4. I think it should be "mean local slope".

**Our reply:** We think "local slope" is correct here.

**References**

Crucifix, M., Does the last glacial maximum constrain climate sensitivity?, *Geophysical Research Letters*, *33*, L18,701, doi: 10.1029/2006GL027,137, *Crucifix* (2006); Crucifix2006; Peter; gotit; awi; AGCM; radiative forcing; LGM; today; 2xCO2; modelling; intercomparsion, 2006.
Dufresne, J.-L., and S. Bony, An assessment of the primary sources of spread of global warming estimates from coupled atmosphere-ocean models, *Journal of Climate*, *21*, 5135–5144, doi: 10.1175/2008JCLI2239.1, *Dufresne and Bony* (2008); DufresneBony2008; Peter; gotit; awi; today; review; RF; feedback analysis; IPCC AR4, 2008.

Hargreaves, J. C., A. Abe-Ouchi, and J. D. Annan, Linking glacial and future climates through an ensemble of gcm simulations, *Climate of the Past*, *3*, 77–87, *Hargreaves et al.* (2007); HargreavesAbe-OuchiAnnan2007; Peter; gotit; awi; modelling; paleo; radiative forcing; zu Masson-DelmotteKageyamaBraconnot2006, 2007.

Köhler, P., B. de Boer, A. S. von der Heydt, L. S. Stap, and R. S. W. van de Wal, On the state dependency of equilibrium climate sensitivity during the last 5 million years, *Climate of the Past*, *11*, 1801–1823, doi:10.5194/cp-11-1801-2015, 2015.

Martínez-Botí, M. A., G. L. Foster, T. B. Chalk, E. J. Rohling, P. F. Sexton, D. J. Lunt, R. D. Pancost, M. P. S. Badger, and D. N. Schmidt, Plio-pleistocene climate sensitivity evaluated using high-resolution $CO_2$ records, *Nature*, *518*, 49–54, doi:10.1038/nature14145, 2015.

PALAEOSENS-Project Members, Making sense of palaeoclimate sensitivity, *Nature*, *491*, 683–691, doi:10.1038/nature11574, 2012.

Rohling, E. J., M. Medina-Elizalde, J. G. Shepherd, M. Siddall, and J. D. Stanford, Sea surface and high-latitude temperature sensitivity to radiative forcing of climate over several glacial cycles, *Journal of Climate*, *25*, 1635–1656, doi:10.1175/2011JCLI4078.1, 2012.

von der Heydt, A. S., P. Köhler, R. S. van de Wal, and H. A. Dijkstra, On the state dependency of fast feedback processes in (paleo) climate sensitivity, *Geophysical Research Letters*, *41*, 6484–6492, doi:10.1002/2014GL061121, 2014.

Yoshimori, M., Hargreaves, J. C., Annan, J. D., Yokohata, T., and Abe-Ouchi, A.: Dependency of Feedbacks on Forcing and Climate State in Physics Parameter Ensembles, Journal of Climate, 24, 6440–6455, doi:10.1175/2011JCLI3954.1, 2011.

---

## Author Comment (AC2) · 18 May 2016

**Response to Reviewer #3**                                              May 19, 2016

3.1 This paper appears to be a comment on, or clarification of, the methods used
in Kohler et al 2015. It is not clearly written and this has made the reviewing a
bit tricky. I had to go back and read Kohler et al 2015 to have a clue what this
is about. It transpires that Kohler et al 2015 and von de Heydt et al 2014 use
different methods to calculate the climate sensitivity, and I think the point of this
manuscript could be to highlight the differences in the results obtained by using
the different methods. But I'm not quite sure. It is stated in the manuscript that
von de Heydt et al used Approach II, but the authors do not clearly state that
Kohler et al used Approach I. Although this manuscript is underwhelming, I think
it may help avoid future problems and confusions by explicitly pointing out the
differences in the two methods. It is a shame that this work was not included as
an appendix to Kohler et al 2015!

**Our reply:** This paper is an extension of what is shown in *Köhler et al.* (2015),
it is not a comment on, or clarification of, the methods used in the 2015 paper.
When we finalized *Köhler et al.* (2015) we (a) have not had all the analysis ready
that we presented here, (b) wanted to focus if there is and how to find state-
dependency. We agree, that it would have been nice to have it included in 2015
already, however, it would have made that paper also a bit more complicated.
With respect to where the approaches are used: Indeed in *von der Heydt et al.*
(2014) approach II has been used, but in *Köhler et al.* (2015) not the full scope of
approach I has been used, but only the subsection, in which a probability density
function (PDF) has been calculated out of individual data points. The reason
why the full use of the equations of approach I was not yet included in *Köhler
et al.* (2015) was the fact that we were aware that approach I and approach II
disagreed, but we had not yet resolved how they might be related to each other
(subsection 2.3 here). In the revised manuscript we will make it clearer to the
reader which approach has been used previously already.

3.2 So of course I had to go back to Kohler et al 2015. In that paper, curves are fitted to temperature / radiative forcing (hereafter RT) plots. They conclude that, in some cases, nonlinear fits may be appropriate and thus climate sensitivity is state dependent. There seem to be two problems with this conclusion. The first is the statement in the introduction of Kohler et al 2015, "However, we are not aware that a difference in the response has been shown for radiative forcing from surface albedo changes ( R[LI]) and CO2 ( R[CO2 ] ). Hence we combine them linearly." A different response to these two forcings was already clearly shown in Yoshimori et al (doi:10.1175/2011JCLI3954.1, 2011). Thus, since the RT curves for "LI" and "CO2" are likely to be different, finding that the RT curves when CO2 and LI forcings are both included is nonlinear does not uniquely show state dependence of climate sensitivity. Rather it more likely shows a combination of state dependence and forcing dependence.

**Our reply:** Thanks for this clarification and of what has been already shown in *Yoshimori et al.* (2011). We will briefly mention this aspect in the draft now. However, this comment is more related to the 2015 paper and even when it would imply that what we show is more likely a combination of state dependence and forcing dependence (and not state dependence alone) of climate sensitivity all aspects how to calculate $S$ out of the scattered data of $\Delta T$ and $\Delta R$ which is show here, is still important.

3.3 The second potential problem in Kohler et al 2015 that is pertinent to the manuscript under review is possible over-fitting of high order polynomials. More parameters means it is easier to fit a scatter of points, and the method used to discriminate between the polynomials should take this into account. I'm not sure exactly how the authors employed the F-test, but why did they not use something

like BIC (Bayesian Information Criterion) which explicitly takes into account this over-fitting problem? It seems likely that the higher order polynomials are not supported by the data. Since the authors are in the mode of commenting on their own previous work, perhaps they could also address this over-fitting issue in the manuscript under review.

**Our reply:** Interestingly, when preparing the paper which was now published in 2015 we first only investigated if a 2nd order polynomial would better fit the data than a linear approach. However, one of the first reviewer, who commented on that manuscript, even before it was submitted to *Climate of the Past*, suggested that we should test if even higher order polynomials might better fit to the data. We therefore followed this idea ever since and tested which order of polynomial best fits the data. For that aim we used 2 different approaches, one based on Akaike's information criterion (AIC), (e.g. see *Wilks*, 2006), the other based on F-test. However, we found that the F-test is more conservative than AIC, implying that when relying on F-test the order of the polynomial was for some cases smaller than when relying on AIC. We therefore restricted all statistics to F-tests only. Since we used two different statistical methods and finally used only the more conservative one we think we are not overfitting the data. This will be mentioned now briefly.

3.4 The main thrust of the manuscript under review is that, in the case of a nonlinear polynomial in RT space, a different result for calculation of the gradient of a curve will be obtained depending on which position on the polynomial you start from. This is obvious. However, the point that I think the authors are making is that Kohler et al 2015 calculated all their values of R relative to a particular state, (Ro,To) and then calculate Sensitivity (S) as (T-To)/(R-Ro) whereas Von de Heydt et al calculate the tangent to the RT curve. These two methods result in a different function for S. The authors state that Approach I (Kohler et al 2015) is the most

robust approach, but it is not clear why, and it seems to me that representing S as dT/dR (Approach II of von der Heydt et al) is more generally our scientific goal. The authors state that Approach II "is more readily comparable to climate model results". This is an odd statement as it really depends on the climate model experiment. It would be possible to exactly reconstruct Approach I using a climate model, and indeed a common suite of experiments (0.5CO2, 2xCO2, 4xCO2 starting from try control state), are a version of Approach I.

**Our reply:** Approach I was not in full depth followed on in *Köhler et al.* (2015) (see our reply to comment 1 of the reviewer above). Approach I was in our view the more obvious or most robust one because the results following this approach are in agreement with the climate sensitivities $S$ calculated for each individual data point (one time step), while results based in approach II disagree with results based in individual points and only converge to those based on approach I when the intergral given in Eq 11 is calculated (see Fig 3b of manuscript under discussion). The important point is that when following the local slope (approach II) to calculate $S$ one needs to calculate the integral given in Eq 11 ($S^{\text{model}} = \frac{1}{\Delta R_2 - \Delta R_1} \int_{\Delta R_1}^{\Delta R_2} S^{\text{local}} \, \mathrm{d}\Delta R$,), and not stop at Eq 5 ($S^{\text{local}} = \frac{\delta \Delta T}{\delta \Delta R}$). The reviewer is right that simulations can be designed which follow approach I, therefore the comment "that Approach II is more readily comparable to climate model results" will be deleted.

3.5 I don't understand the point of the Discussion section. It isn't a discussion of the previous sections, but another comment on a different part of the analysis. It seems to be comparing two completely different things. One thing is the fitting of a curve to the scatter of points in RT space, which results in the calculation of a functional relationship between S and T. The second thing is taking the scatter of points in the RT space and turning them into a distribution in R (or S) space, which indicates how often the earth system has wandered into different parts of

RT space in the paleoclimate record.

**Our reply:** What previously has been the discussion section is either shifted to other sections or deleted, because being a rather inimportant statement. See also comment 5 of reviewer 1 and our reply to it, pointing in the same direction.

**References**

Köhler, P., B. de Boer, A. S. von der Heydt, L. S. Stap, and R. S. W. van de Wal, On the state dependency of equilibrium climate sensitivity during the last 5 million years, *Climate of the Past*, *11*, 1801–1823, doi:10.5194/cp-11-1801-2015, 2015.

von der Heydt, A. S., P. Köhler, R. S. van de Wal, and H. A. Dijkstra, On the state dependency of fast feedback processes in (paleo) climate sensitivity, *Geophysical Research Letters*, *41*, 6484–6492, doi:10.1002/2014GL061121, 2014.

Wilks, D., *Statistical methods in the atmospheric sciences, second edition*, *International Geosphere Series*, vol. 59, Academic Press, 2006.

Yoshimori, M., J. C. Hargreaves, J. D. Annan, T. Yokohata, and A. Abe-Ouchi, Dependency of feedbacks on forcing and climate state in physics parameter ensembles, *Journal of Climate*, *24*(24), 6440–6455, doi:10.1175/2011JCLI3954.1, 2011.

---

## Author Comment (AC3) · 18 May 2016

**Response to Reviewer #2 (Jonah Bloch-Johnson)**  May 19, 2016

**Our reply:** This rather long review only deals with the question if in the scatter plot of $\Delta$R-$\Delta$T from which climate sensitivity might be calculated, $\Delta$R is the independent variable (plotted on the x-axis) and $\Delta$T is the dependent variable (plotted in the y-axis) or vice versa. The reviewer was already a reviewer of our 2015 paper and this point was already brought up in the discussion phase in 2015. Accordingly, we tested the importance of this assumptions on our results and layed out some arguments why we think $\Delta$R should stay on the x-axis. Please refer to the peer-review process of the 2015 paper for details. One of our arguments was and still is, that the calculation of $S$ from palaeodata developed around the definition we put forward in Eq 1: $S = \Delta T / \Delta R$, from which the most natural choice of calculating $S$ as the slope of the regression line in the linear case has developed. This is and was our starting point, from which we developed how the analysis might be performed, once the system is not linear anymore.

Reading this review and the own paper of the reviewer (*Bloch-Johnson et al.*, 2015) in detail, we think we see the reason why he insists on switching the axes. This might come from the term equilibrium non-linearity as defined in that paper. It is based on the generally used Taylor-expansion of $\Delta R = \lambda \Delta T + a(\Delta T)^2 + O((\Delta T)^3)$. Normally, and of course in the whole concept of $S$ this expansion is simply taken as first order, so $S = -1/\lambda = \Delta T / \Delta R$. In that paper (*Bloch-Johnson et al.*, 2015), the authors show that there may be quadratic or even higher order corrections to this expansion that become particularly important in the long tails of climate sensitivity distributions. In that sense, writing $\Delta R = \lambda \Delta T + a(\Delta T^2)$, $\Delta R$ is a function of $\Delta T$ and should be plotted on the y-axis (with $\Delta T$ on the x-axis).

We argue differently: We say that $\Delta T = S \Delta R$ and $S$ might not be constant, but a (unknown) function of $T$ and $R$. It is, therefore, not even clear that there exists a functional relation $T(R)$ or $R(T)$, and we try different approaches to directly estimate $S$ from the cloud of points. In this case it doesn't matter at all whether we use $T - R$ or $R - T$ relations.

Some comments on the details given in this review:

- In the introduction some fundamental motivation why temperature should be the independent variable are given. Nothing to be added there, apart from the fact, that for palaeostudies the forcings and feedbacks that change the climate system with respect to pre-industrial climate, are the interesting parts, and that models typically calculated temperature changes as response to those (making temperature the dependent variable again).

- In section 2.1 it is argued that temperature change as function of forcing change is not necessarily a well-defined function, bringing some arguments about bifurcation and a reanalysis of some simple model-based calculations some of us have published in *von der Heydt et al.* (2014). Note, that this model is unrealistically simple and was used only to verify the difference between fast and slow feedback processes. Furthermore, even in *von der Heydt et al.* (2014) we put forward an analysis in a scatter plot, similar to here, with $\Delta R$ being on the x-axis.

- In section 2.3 the non-linear regressions shown in 2015 are repeated, but with x and y changed in the analysis to determine a non-linear function. It is argued, that now even the fit through the data based on $CO_2$ from the Pagani and Foster lab (apart from one case) agree with our main finding, that climate sensitivity is lower in colder periods. So far, we did not find any significant non-linearity in the Foster and Pagani data sets. We think, that this support found for our earlier findings is encouraging, but the fits through both Foster and Pagani for both our approach ($\Delta R$ on x-axis) and the approach of the reviewer ($\Delta T$ on x-axis) is far away from going through the origin ($\Delta T=0$ when $\Delta R = 0$). While in our view meeting the origin is of fundamental importance in this overall approach, we think these new results are therefore only of limited usage and will not change the conclusions we put forward in 2015.

In principle we can follow the (physically motivated) arguments put forward why we should switch x and y, however, we can not see, that this should be uttermost important (see above our understand why we think the reviewer insists on switching the axis). Saying that, we agree that there might be benefits in following this switch in the axes, but we can not say that it will change the overall findings. The core of this review is on

the method of how to set up the scatter plot. Thus, it is more on how good (or bad) the previous papers we published have been setup. Interestingly, the reviewer was finally happy with our setup in 2015, and did not insist on change axes in the scatter plots, while he here seemed to rethink his former position.

How to continue? The issues we bring up (the quantification of $S$ based on local slope versus pointwise, how they disagree and how they can be transformed into each other) would also show up when we switch axes, the equations how to transform one into the other might in detail be different. Therefore we think this technical note is well served if, for the time being, it is performed in the setup as is. However, we keep the suggested switches of axis in mind and might continue future investigations in that direction following the suggestions brought up here.

**References**

Bloch-Johnson, J., R. T. Pierrehumbert, and D. S. Abbot, Feedback temperature dependence determines the risk of high warming, *Geophysical Research Letters*, *42*(12), 4973–4980, doi:10.1002/2015GL064240, 2015GL064240, 2015.

von der Heydt, A. S., P. Köhler, R. S. van de Wal, and H. A. Dijkstra, On the state dependency of fast feedback processes in (paleo) climate sensitivity, *Geophysical Research Letters*, *41*, 6484–6492, doi:10.1002/2014GL061121, 2014.

---

## Author Comment (AC4) · 18 May 2016

See our reply to the other comment of reviewer #2 in which we accidentially uploaded all the detailed replies that should show up here. (called AC3: 'Rebuttal to reviewer #2' in the discussion).